# Biotic Integrity, Water Quality, and Landscape Characteristics of a Subtropical River

**Luis Fernando Gudiño-Sosa** [1], **Rodrigo Moncayo-Estrada** [2,*], **Martha Alicia Velázquez-Machuca** [1,*],
**Gustavo Cruz-Cárdenas** [1], **Luis Arturo Ávila-Meléndez** [1] and **José Luis Pimentel-Equihua** [3]

[1] Instituto Politécnico Nacional, Centro Interdisciplinario de Investigación para el Desarrollo Integral Regional, Unidad Michoacán, Justo Sierra No. 28, Jiquilpan 59510, Michoacán, Mexico; fherr0322@gmail.com (L.F.G.-S.); guscruz@ipn.mx (G.C.-C.); lavilam@ipn.mx (L.A.Á.-M.)

[2] Instituto Politécnico Nacional, Centro Interdisciplinario de Ciencias Marinas, Av. Instituto Politécnico Nacional s/n Col. Playa Palo de Santa Rita, Apdo. Postal 592, La Paz 23096, Baja California Sur, Mexico

[3] Colegio de Postgraduados-Campus Montecillo, Carretera Federal México-Texcoco Km 36.5, Montecillo 56230, Estado de México, Mexico; jequihua@colpos.mx

* Correspondence: rmoncayo@ipn.mx (R.M.-E.); mvelazquezm@ipn.mx (M.A.V.-M.)

**Abstract:** The integrity of rivers is affected by anthropogenic activities at different spatial scales, from basin and landscape levels to the direct effects on the river and aquatic life. Our objective was to study these effects on the subtropical La Pasión River, analyzing environmental, geomorphological, habitat and water quality, and macroinvertebrates. We sampled the dry season (March 2022) because the river presented stable conditions. We selected the most influential variables in each spatial scale and determined their relationship with the indexes of quality characteristics and aquatic life in the river using multivariate statistics. Most sites (≈65%) had medium water and suboptimal habitat quality status, meanwhile half the sites had regular biotic integrity status; without finding coincidence in the quality of the different indexes applied, all sites indicated a high gradient of degradation from the origin to the mouth of the river. The presence of some families (e.g., Culicidae, Chironomidae, Lumbriculidae) indicated organic matter contamination. The main variables that significantly classified the river quality and integrity structure were water flow, turbidity, habitat embeddedness, and sulfates ($\chi^2 = 0.1145$, $p < 0.01$). It is concluded that the affected sites received wastewater without prior treatment and presented physical barriers such as irrigation channels.

**Keywords:** water quality index; BMWP; Hill's numbers; multivariate analysis; GIS

## 1. Introduction

Due to a continuous and growing process of pollution and degradation, rivers stand out as the aquatic ecosystems most affected by anthropogenic activities [1–3]. Therefore, there is a loss of environmental services and biodiversity [4–6]. In addition, the water quality of the river depends on the ecosystem's interaction with its surroundings, both at the landscape and basin levels. Changes in land use and productive activities (e.g., livestock, dairy industry, and agriculture) promote pollution and habitat deterioration and the decrease of resource availability. Additionally, the development of urban and rural areas where a wide variety of activities are carried out, including industrial ones, impacts rivers due to mixed untreated wastewater discharges [7,8].

In this context, it is essential to have different ways to identify, quantify, and value the quality of the resource, the habitats, and their biotic integrity. On the one hand, monitoring has been implemented through physicochemical variables and the application of water quality indices to understand the characteristics and condition of the resource mathematically [9,10]. On the other hand, there is biomonitoring, in which aquatic organisms are used as a surrogate to understand the water and habitat quality [11]. Biotic integrity indices,

such as the Biological Monitoring Working Party index (BMWP) and other ecological indices that work with macroinvertebrates have been used as descriptive tools [12]. Aquatic macroinvertebrates are used as bioindicators because of their ability to reflect the real conditions of the water bodies, habitat quality, channel alterations, hydromorphological characteristics, ecosystem quality, and ecosystem services.

In this study we selected the subtropical La Pasión river because it presents a distinctive punctual and diffuse pollution from the interaction of productive activities, urban and industrial development, and continuous change in land use. Furthermore, this river is an important tributary for Lake Chapala, which is the largest natural lake in Mexico and a RAMSAR site (n° 1973 with a surface of 114,659 ha). The main aim was to analyze the impacts presented from the riverbed to the landscape and basin levels in an integral manner, using both abiotic (water and habitat quality) and biotic (biological integrity and indicator species) indices. Moreover, we implemented a protocol of different statistical analysis to better describe the community and its relationship with the environment. We hypothesized, first, that a greater impact on water quality and the macroinvertebrate community is expected for the middle part of the river due to activities related to agriculture and the dairy industry and that towards the end of the river there will be impacts on the quality of the habitat due to urban development, which will generate a gradient from better (origin) to worse (river mouth) in terms of water quality and loss of biodiversity [13]. Second, the impacts are directly reflected successively from a larger scale (basin) to local characteristics within the river channel (water and habitat quality and macroinvertebrates).

## 2. Materials and Methods

### 2.1. Study Area

The La Pasión river basin is located within parallels 19°57′ and 20°11′ N and the meridians 102°51′ and 103°12′ W in Mexico, within an altitude that oscillates between 1500 and 2400 m a.s.l., and has an area of 560,909 km². The river is considered one of the main currents in the basin, with a length of 29.56 linear km. The La Pasión river runs through three municipalities, Marcos Castellanos, Tizapán el Alto, and La Manzanilla de la Paz, and flows into the south shore of Lake Chapala [14]. The basin is located within the Lerma-Chapala-Pacífico administrative region, in the Lerma hydrological region number 7 or Lerma-Chapala sub-basin (Figure 1). We defined the basin with the QGIS program (QGIS Development Team, 2021) [15] by applying four algorithms to a digital elevation model obtained from the geographic information systems platform at INEGI [16]: (1) fill sinks, (2) Strahler order, (3) defining channel networks, and (4) calculating upslope and downslope areas [17,18].

### 2.2. Fieldwork and Sample Analysis

Eighteen sites with a regular distribution were sampled on the main course of the river and on adjacent channels, including natural features (e.g., tributaries inflow and river mouth), anthropogenic impacts (e.g., domestic and industrial wastewater inputs), and reference sites with little impacted conditions to make comparisons among different water qualities [19]. Sampling was carried out in the dry season (March 2022) since it represented the low flow phase in the river when macroinvertebrate sampling is most effective. There is also hydrological connectivity between the sampling sites in the river because, at the peak of the dry season (end of May and early June), intensive water use for irrigation can isolate parts of the main channel [20]. Finally, human impacts are magnified by low flows, creating spatial variation throughout the river system that reflects organism responses to different stressors [21–23].

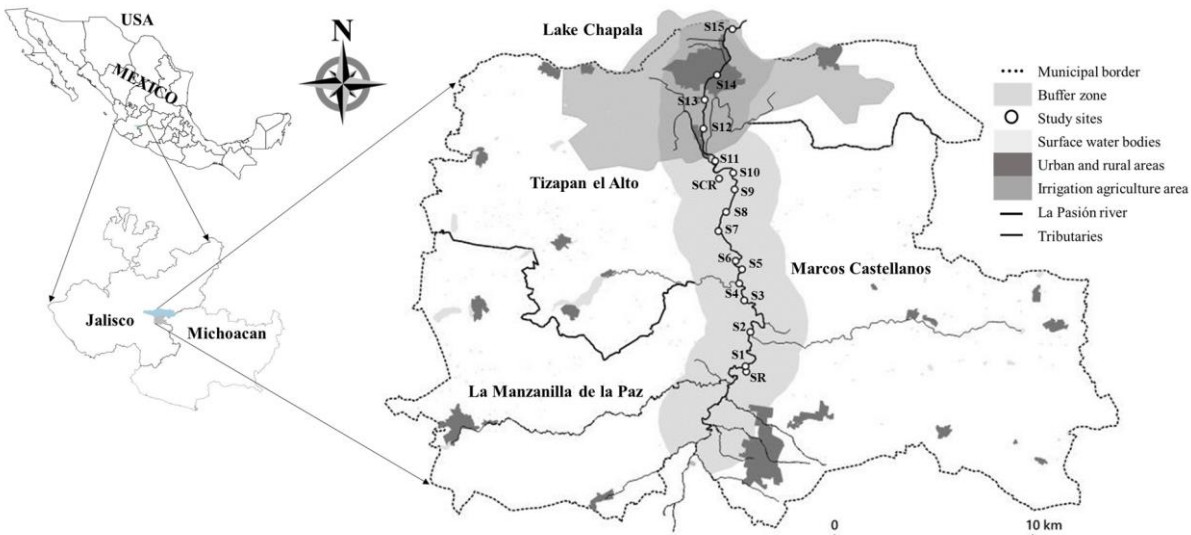

**Figure 1.** Location and sample sites (SR, SCR, and S1 to S15) in the La Pasión River basin, Mexico. Lake Chapala, the largest in the country, is represented in blue.

Water physicochemical parameters such as temperature, percentage of dissolved oxygen saturation, pH, electrical conductivity, and salinity were recorded at each sampling site with a multisensor YSI Pro1030 (YSI Incorporated, Yellow Springs, OH, USA). Transparency was measured with a Secchi disk and the discharge with a Flowatch flowmeter AMI0608 (JDC Electronic SA, Yverdon-les-Bains, Switzerland). In addition, water samples were collected at each site in wide-mouth amber glass bottles of 500 mL capacity and amber polypropylene bottles of 1000 mL capacity. In the laboratory, oils and grease (O and G in mg/L) were determined by Soxhlet extraction [24], and microbiological analyzes for total and fecal coliforms and *Escherichia coli* were determined by the most probable number technique (MPN/100 mL) [25]. Likewise, acidity–alkalinity ($CO_3$ and $HCO_3$ in mg/L), hardness ($CaCO_3$ in mg/L), chlorides ($Cl^-$ in mg/L), boron (B in mg/L), chemical and biochemical oxygen demand (micro COD and BOD5 in mg/L), total phosphate ($PO_4^{3-}$; TP in mg/L), nitrogenous compounds such as ammoniacal nitrogen, nitrates, and nitrites ($NH_4$, $NO_3$, and $NO_2$ in mg/L), and sulphates ($SO_4^{2-}$ in mg/L) were all determined with the methodologies proposed by APHA [22] and current Mexican Norms. Major cations ($Ca^{2+}$, $K^+$, $Mg^{2+}$ and $Na^+$) and total dissolved metals (Cd, Cr, Cu, Fe, Mn, Ni, Pb and Zn) were measured with atomic absorption spectrometry [26].

The aquatic macroinvertebrates were captured with a Surber-type net (mesh size of 500 μm) within a 100 m section of the river at each site [23,27]. We sampled different hydromorphological units (pools, riffles, and rapids) and habitats (floating material, aquatic and riparian plants, sediment, and other bed substrates) [14,27,28]. The organisms were preserved in 2 L bottles with 70% alcohol. The samples were washed in a bucket with a 420 μm sieve bottom to remove most of the clay and silt. Aquatic organisms were separated by density difference in a supersaturated sugar solution, recovering the supernatant with a 420 μm mesh opening net for manual separation, quantification, and classification with the help of a stereomicroscope, dichotomous keys, and a specialized bibliography [22,29,30].

### 2.3. Hydrogeomorphological Characteristics and Indices

We analyzed the landscape characteristics by establishing an area of 2 km × 2 km (buffer zone) around each sampling site (the site in the middle point) [31]. The land use and vegetation in the riverbanks were described from the land use series maps of the National Institute of Statistical and Geography at a scale of 1:250,000 (INEGI; shapefiles format). We characterized the habitat quality describing the 10 parameters related to the riverbed and riverbank following the protocol proposed by Cornejo et al. [32]. Additionally, the geomorphological characteristics of the river were defined with the classification proposed

by Rosgen [33]. We quantified the land use and vegetation cover using the supervised classification of satellite images using the machine learning regression algorithm (MLR) with a *p*-value of 0.8814 and kappa index of 0.783 (Lansat8 OLI TIRS from the USGS platform) [34,35]). The normalized difference vegetation index (NDVI) [21,36] was applied to discover the state of the vegetation surrounding the river, and the erosion index (E), through the universal soil loss equation [37], was used to determine the degree of erosivity. Finally, data regarding the main productive activities in the municipalities was obtained from the National Institute of Statistical and Geography municipal records [38].

### 2.4. Abiotic and Biotic Indices

We applied the National Sanitation Foundation index of water quality (NSF-WQI; https://www.nsf.org (accessed on 20 March 2022) [11,39–41], which includes nine physico-chemical parameters (dissolved oxygen, fecal coliform, pH, BOD5, temperature change, total phosphate, nitrate, turbidity, and total solids), to assess the quality of the water. All parameters are classified in a water quality range and multiplied by a weighting factor, and the results are classified into five categories from very bad to excellent on a scale of 0 to 100 [40]. The diversity of macroinvertebrates was evaluated among sites by implementing rarefaction to standardize the odd number of samples and extrapolation to predict the actual diversity with respect to the expected number of species not obtained within the sampling effort [42]. Hill's number (three Hill number) is proposed by [43] and based on both methods: q = 0 (species richness), q = 1 (Shannon–Wiener index), and q = 2 (inverse Simpson index). Other biological indices selected for the analysis were the percentage of Ephemeroptera (minus Baetidae), Plecoptera, and Trichoptera (EPT-B %) [44] and the Biomonitoring Working Party (BMWP) index adapted for subtropical rivers, which considered the sensitivity or tolerance to pollution of macroinvertebrate families [45,46].

### 2.5. Statistical Analyses

We screened the abiotic variables for redundancy with Pearson's correlation to eliminate highly redundant variables (correlations > 0.95) [47]. Moreover, a multi-factor analysis, not presented here, was used to determine those variables that accounted for the greatest proportion of variance (≥0.4 of cos2) [48]. We discarded site 16 (S16) from these analyses because it had outliers in several physicochemical parameters. Since data distribution significantly differed from normality (Shapiro test), the non-parametric Kruskal–Wallis analysis of variance with the Dunn test (as an posteriori test) was adopted to test for differences between densities in the sampling sites.

We used two analyses to describe the macroinvertebrate community across the different sets of environmental variables. First, we applied a non-metric multidimensional scaling (NMDS) with a Bray–Curtis dissimilarity matrix to examine variation in the families [49,50]. A three-dimensional solution was calculated using 250 random starts of real data and 1000 iterations to evaluate stability (final stress of 0.08). We integrated the environmental variables as vectors in the ordination plot; this information is not part of the NMDS analysis (vectors are scaled by their correlation with the axes). Secondly, we used a multivariate regression tree analysis (MRT) [51–53] based on the Bray–Curtis dissimilarity matrix. This is a prediction model to examine patterns between the variation of the macroinvertebrate's abundance and biotic indices as dependent variables and the different independent quantitative environmental variables (there are 48 including water quality, basin, geomorphology, and habitat). The MRT helps to identify the variable interactions because the information is partitioned into smaller sections and creates a tree of dichotomies. Each dichotomy minimizes the dissimilarity of samples within each tree branch. The tree with the lowest cross-validated relative error was reported according to the 1-SE rule [54]. Kruskal–Wallis tests were performed with the 'dunn.test' package (v. 1.3.5) [55], NMDS was computed using the vegan package (v. 2.5-7) [56], and the MRT was calculated using the 'mvpart' package (v. 1.6-2) [57], all in the R language [58].

## 3. Results

### 3.1. Landscape Characteristics

Regarding landscape characteristics, semideciduous forest, grassland, and agriculture showed higher surface usage, whereas human settlements covered less of the land (Table 1). Sites located in the lower basin had more human settlement coverage and S14 presented 100% coverage because it belonged to an urban park in the Tizapan town. The agricultural use was also located in the lower zone of the fluvial landscape close to the outflow, and five sites had coverages greater than 50%. The main crops produced were corn, green beans, sorghum, onion, cabbage, broccoli, oats, peas, tomato, squash (pumpkin), and berries. Induced grassland was located mainly in the upper zone of the fluvial landscape, and four sites showed coverage values greater than 50%. The lowland semi-deciduous forest was the most representative cover in six sites in the middle of the fluvial landscape, with values from 40 to 98%. This vegetation also had some important species of columnar cacti (Table 1).

**Table 1.** Main land use and vegetation cover ($km^2$) in the different sites of the La Pasión river fluvial landscape.

| Sites | Human Settlements | Agriculture | Grassland | Semideciduous Forest |
|---|---|---|---|---|
| SR | 0 | 0 | 2.19 | 0 |
| S1 | 0 | 0 | 2.19 | 0 |
| S2 | 0 | 0 | 218.53 | 0 |
| S3 | 0 | 0 | 999.74 | 0 |
| S4 | 0 | 0 | 639.02 | 218.68 |
| S5 | 0 | 0 | 639.02 | 218.68 |
| S6 | 0 | 0 | 81.22 | 487.37 |
| S7 | 0 | 0 | 210.17 | 639.19 |
| S8 | 0 | 0 | 114.49 | 885.27 |
| S9 | 0 | 0.07 | 23.67 | 975.27 |
| S10 | 0 | 502.68 | 0 | 409.96 |
| S11 | 92.73 | 626.82 | 0 | 144.46 |
| S12 | 16.95 | 93.07 | 0 | 0.57 |
| S13 | 0 | 543.85 | 0 | 455.83 |
| S14 | 999.77 | 0 | 0 | 0 |
| S15 | 0.08 | 378.54 | 0 | 0 |
| SCR | 16.95 | 93.07 | 0 | 0.57 |
| Total | 1126.48 | 2238.1 | 4109.33 | 2930.24 |

### 3.2. Water and Habitat Characteristics

According to the physicochemical characteristics, some sites presented low variable values with small variances, such as $BOD_5$, $PO_4$, and $SO_4$ in SR and S1, and have better environmental characteristics than sites with greater variability, with values even up to two orders of magnitude and extreme data (e.g., S6, S13, S16) that have punctual impacts on untreated water discharges (Table 2 and the complete variables in Table A1).

Concurrently, the results of the water quality index contrast between sites. There was a maximum value of 86, which means good quality (12%), and values of bad and very bad quality (23%); however, most of the sites were classified as medium quality (65%; Table 3). Regarding the habitat characteristics (habitat quality index, HQI), most of the sites were suboptimal (67%) and the fewest were marginal (11%).

### 3.3. Macroinvertebrates Characteristics

The diversity of aquatic macroinvertebrates consisted of a total of 14,829 individuals, represented by 67 families classified into 24 orders, 10 classes, and 5 phyla. The most frequent and abundant aquatic macroinvertebrates were Culicidae (27.4%), Chironomidae (15.11%), Lumbriculidae (9.05%), Baetidae (5.72%), Polycentropodidae (4.89%), and Physidae (3.27%) (Table 4); the remaining types accounted for 35% of the total.

**Table 2.** Most important physicochemical variables influencing the water quality index and the habitat quality index in the different sites of the La Pasión river.

| Sites | pH | CO$_3$ (mg/L) | HCO$_3$ (mg/L) | SO$_4^{2-}$ (mg/L) | PO$_4^{3-}$-TP (mg/L) | NH$_4$ (mg/L) | NO$_3$ (mg/L) | COD (mg/L) | BOD$_5$ (mg/L) | Turb (NTU) | Flow (m$^3$/s) |
|---|---|---|---|---|---|---|---|---|---|---|---|
| S1 | 7.73 | 0.13 | 3.13 | 0.0002 | 0.12 | 4.71 | 0 | 12.33 | 2.12 | 19 | 0.27 |
| S2 | 7.98 | 0 | 4.20 | 0.011 | 1.23 | 1.17 | 0.04 | 15.67 | 2.7 | 65 | 0.64 |
| S3 | 8.24 | 0 | 4.85 | 0.008 | 0.55 | 1.42 | 0.32 | 149 | 27.21 | 100 | 0.69 |
| S4 | 6.87 | 0 | 9.51 | 0.073 | 25.87 | 2.38 | 2.16 | 1065.67 | 477.19 | 50 | 0.70 |
| S5 | 7.68 | 0 | 4.35 | 0.047 | 0.18 | 1.25 | 0 | 42.33 | 369.05 | 50 | 1.19 |
| S6 | 7.18 | 0 | 3.62 | 0.006 | 3.65 | 11.57 | 0.10 | 12.33 | 188.54 | 40 | 0.49 |
| S7 | 7.64 | 0 | 3.49 | 0.0003 | 0.00 | 2.39 | 0.10 | 159 | 27.4 | 35 | 0.69 |
| S8 | 7.65 | 0 | 3.54 | 0.002 | 3.33 | 2.04 | 0 | 185.67 | 81.9 | 13 | 0.95 |
| S9 | 7.65 | 0 | 3.56 | 0.0018 | 3.01 | 3.25 | 0.10 | 29 | 62.83 | 14 | 0.79 |
| S10 | 7.57 | 0 | 3.56 | 0.004 | 0.37 | 1.88 | 0 | 2.33 | 25.04 | 10 | 1.01 |
| S11 | 8.04 | 0.25 | 3.54 | 0.006 | 2.83 | 3.10 | 0.72 | 12.33 | 107.3 | 11 | 0.69 |
| S12 | 8.08 | 0 | 3.62 | 0.008 | 3.20 | 2.04 | 0.35 | 39 | 596.36 | 9 | 0.30 |
| S13 | 8.49 | 0.56 | 5.18 | 0.012 | 0.12 | 6.59 | 0.10 | 12.33 | 188.54 | 30 | 0.62 |
| S14 | 8.29 | 0.33 | 5.59 | 0.015 | 1.28 | 2.27 | 0.01 | 19 | 165.65 | 24 | 0.51 |
| S15 | 7.71 | 0 | 9.58 | 0.102 | 22.07 | 13.96 | 0.44 | 142.33 | 667.02 | 24 | 0.64 |
| SR | 7.73 | 0.13 | 3.13 | 0.0002 | 0.12 | 4.71 | 0 | 12.33 | 2.12 | 19 | 0.27 |
| SCR | 8.04 | 0.25 | 3.54 | 0.006 | 2.83 | 3.10 | 0.72 | 12.33 | 107.3 | 11 | 6.97 |

**Table 3.** Values of the water quality index (NSF-WQI), the habitat quality index (HQI), the Ephemeroptera, Plecoptera, and Trichoptera index minus Baetidae (EPT-B %), and the BMWP, including their interpretation. SR = Reference site; SCR = Site into a channel.

| Sites | NSF-WQI | Significance | HQI | Significance | EPT% | BMWP | Significance |
|---|---|---|---|---|---|---|---|
| SR | 86 | Good | 179 | Optimum | 88.5 | 269 | Excellent |
| S1 | 86 | Good | 173 | Optimum | 6.2 | 172 | Excellent |
| S2 | 64 | Medium | 163 | Suboptimum | 0 | 68 | Regular |
| S3 | 49 | Bad | 157 | Suboptimum | 0 | 70 | Regular |
| S4 | 48 | Bad | 160 | Suboptimum | 0 | 15 | Very polluted |
| S5 | 56 | Medium | 156 | Suboptimum | 0 | 54 | Polluted |
| S6 | 50 | Medium | 152 | Suboptimum | 107.1 | 139 | Excellent |
| S7 | 60 | Medium | 155 | Suboptimum | 63.9 | 88 | Regular |
| S8 | 57 | Medium | 155 | Suboptimum | 11.1 | 127 | Excellent |
| S9 | 59 | Medium | 167 | Optimum | 6.9 | 106 | Good |
| S10 | 64 | Medium | 172 | Optimum | 5.9 | 154 | Excellent |
| S11 | 59 | Medium | 163 | Suboptimum | 51.5 | 135 | Excellent |
| S12 | 58 | Medium | 152 | Suboptimum | 1.9 | 95 | Regular |
| S13 | 56 | Medium | 127 | Suboptimum | 2.3 | 69 | Regular |
| S14 | 55 | Medium | 140 | Suboptimum | 0 | 67 | Regular |
| S15 | 50 | Medium | 132 | Suboptimum | 0 | 104 | Good |
| S16 | 22 | Very bad | 54 | Marginal | 0 | 22 | Very polluted |
| SCR | 59 | Medium | 74 | Marginal | 6.8 | 71 | Regular |

There was a significant difference in the macroinvertebrate's abundance among sampling sites ($\chi^2 = 146.73$, $p < 2.2 \times 10^{-16}$). For instance, the reference site is significantly different from the rest of the sites and site one was only similar to site 10 (S10), which also had optimum habitat condition and excellent biotic integrity (Table 3). Although site 16 (S16) is highly contaminated, it was similar in terms of dominant species to the other sites. According to the results of the BMWP biological index, the quality of the largest number of sites is regular and 33% of the sites had an excellent level, including the reference sites and at the beginning of the sampling (SR and S1; Table 3).

**Table 4.** The most frequent and abundant aquatic macroinvertebrates in the different sites of the La Pasión river.

| Sites | Culicidae | Chironomidae | Lumbriculidae | Baetidae | Polycentropodidae | Physidae | Asellidae |
|---|---|---|---|---|---|---|---|
| S1 | 10 | 277 | 8 | 15 | 127 | 26 | 17 |
| S2 | 20 | 18 | 15 | 49 | 29 | 16 | 94 |
| S3 | 2230 | 17 | 0 | 0 | 0 | 219 | 0 |
| S4 | 633 | 41 | 1254 | 0 | 0 | 7 | 1 |
| S5 | 786 | 3 | 5 | 0 | 0 | 36 | 0 |
| S6 | 276 | 26 | 27 | 0 | 0 | 130 | 0 |
| S7 | 1 | 59 | 0 | 220 | 1 | 7 | 56 |
| S8 | 0 | 166 | 1 | 41 | 84 | 2 | 8 |
| S9 | 3 | 231 | 20 | 45 | 38 | 7 | 26 |
| S10 | 17 | 533 | 0 | 15 | 39 | 3 | 23 |
| S11 | 6 | 310 | 6 | 30 | 28 | 6 | 41 |
| S12 | 4 | 270 | 6 | 84 | 378 | 13 | 7 |
| S13 | 4 | 41 | 0 | 14 | 1 | 0 | 3 |
| S14 | 18 | 39 | 0 | 26 | 0 | 0 | 0 |
| S15 | 48 | 48 | 0 | 196 | 0 | 0 | 0 |
| SR | 8 | 95 | 0 | 2 | 0 | 10 | 14 |
| SCR | 0 | 67 | 0 | 112 | 1 | 4 | 0 |
| S16 | 15 | 0 | 2 | 0 | 0 | 0 | 0 |

In the description of the biological community, the NMDS analysis showed the relationship between the different environmental variables and the presence of the macroinvertebrate families at different sites. The first axis at the top left shows the sites with the best water quality (higher dissolved oxygen concentration and transparency), some aspects of the basin (higher NDVI and irrigation agriculture), and good geomorphological (lower depth and higher flow) and habitat characteristics (rapids, less channel disturbance, more stable shores) (Figure 2).

The families found usually occur in clean waters with little organic pollution, such as Athericidae, Hydropsychidae, Planariidae, Helycopsychidae, Pyralidae, Calopterygidae, and Thiaridae. On the other hand, to the right, there are higher values for parameters that indicate lower water quality with the presence of organic pollution (oils and grease, biochemical oxygen demand, total phosphate, $NO_3$, $NH_4$, and *E. coli*), there is more grassland induced, and a lower habitat quality. In these sites, we capture tolerant families such as Syrphidae, Culicidae, Lumbriculidae, Nematoda, Physidae, Dixidae, and Chironomidae. In the lower part of the ordination space, the vectors were related to higher values for total Coliforms and carbonates, as well as the deepest and widest part of the river, and a large part of the urban settlements and irrigated agriculture in addition to a lower slope. Close to the mouth of the river, we found detritivorous families such as Gammaridae and Muscidae.

The MRT related the macroinvertebrate diversity (Families) to physicochemical variables and hydromorphological and habitat characteristics (Figure 3). The main divisions were according to turbidity, channel alteration, and embeddedness, as shown in three groups and two classification nodes and supported by a significant outcome of the $\chi^2$-test ($\chi^2 = 0.1145$, $p < 0.01$, based on 100 multiple cross-validations).

When we used the biotic indices as dependent variables, sulphates were the most important aspect to separate the sites as well as flow and $Ca^{2+}$. The BMWP index was the most representative and a better indicator of the different splits, as well as the % EPT when lower $SO_4^{2-}$ values were found (Figure 4), as is shown in three groups and two classification nodes and supported by a significant outcome of the $\chi^2$-test ($\chi^2 = 0.2648$, $p < 0.01$, based on 100 multiple cross-validations).

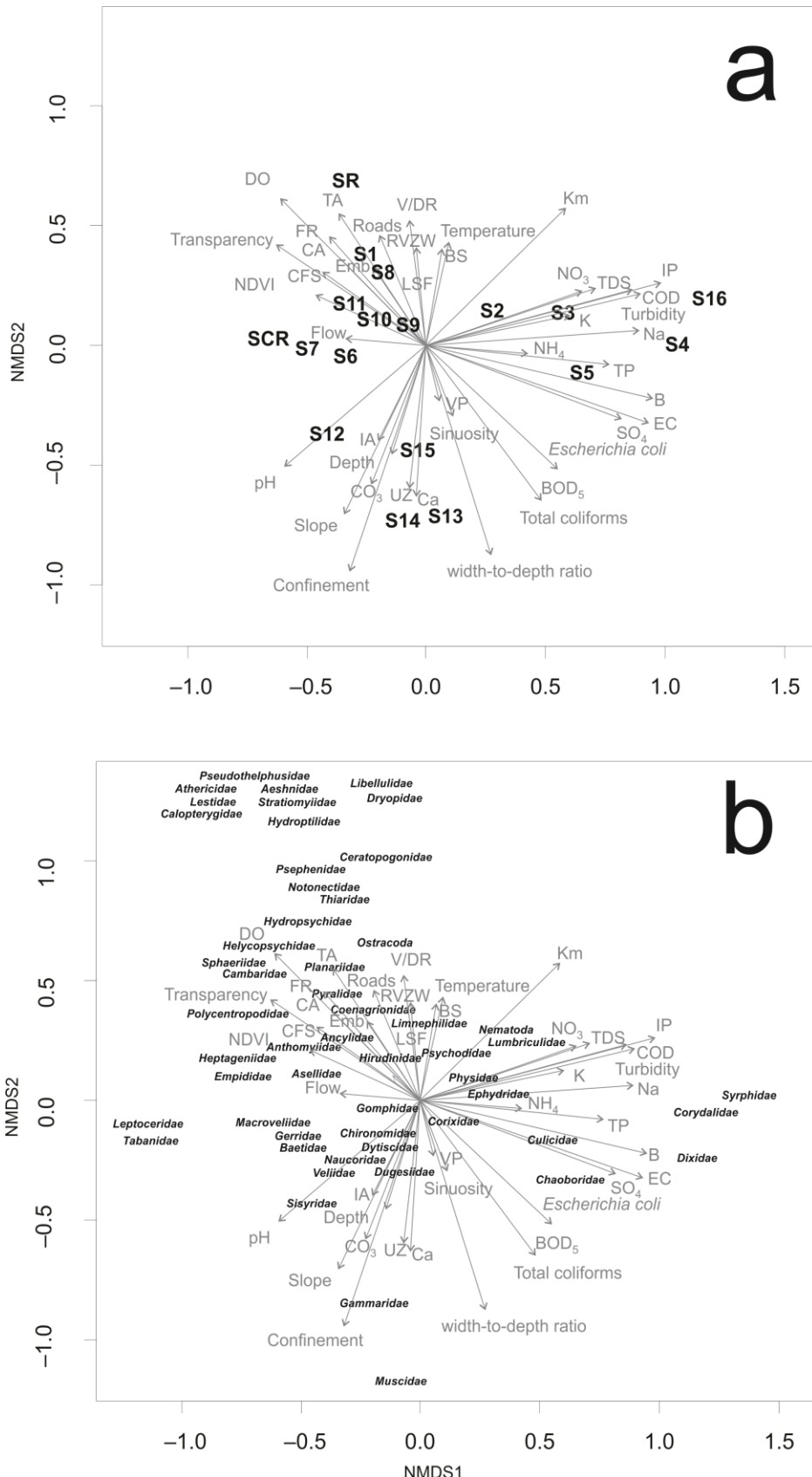

**Figure 2.** Non-metric multidimensional scaling of (**a**) sites and (**b**) macroinvertebrate families. Different environmental variables were included as vectors.

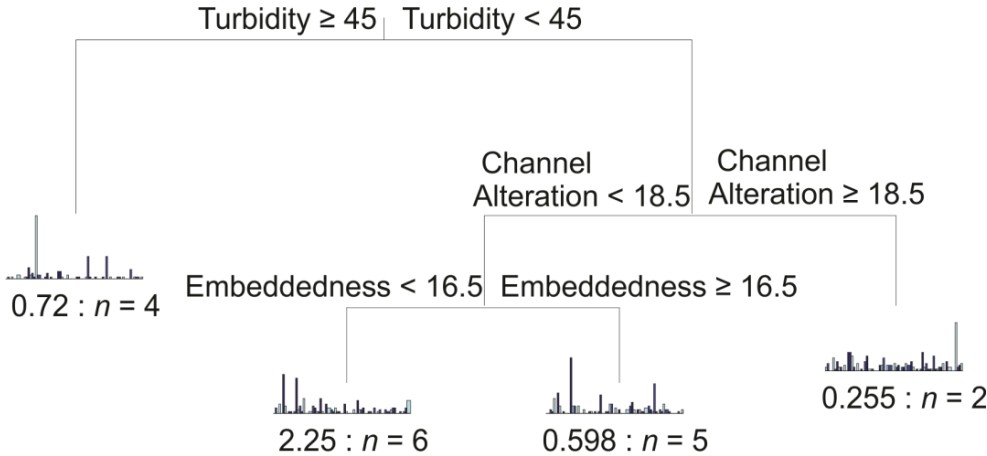

**Figure 3.** Multivariate regression tree (MRT) for the abundance data of macroinvertebrates, using 48 different explanatory variables related to water quality, basin aspects, geomorphology, and habitat quality (<less than; ≥greater than or equal; *n* number of sites).

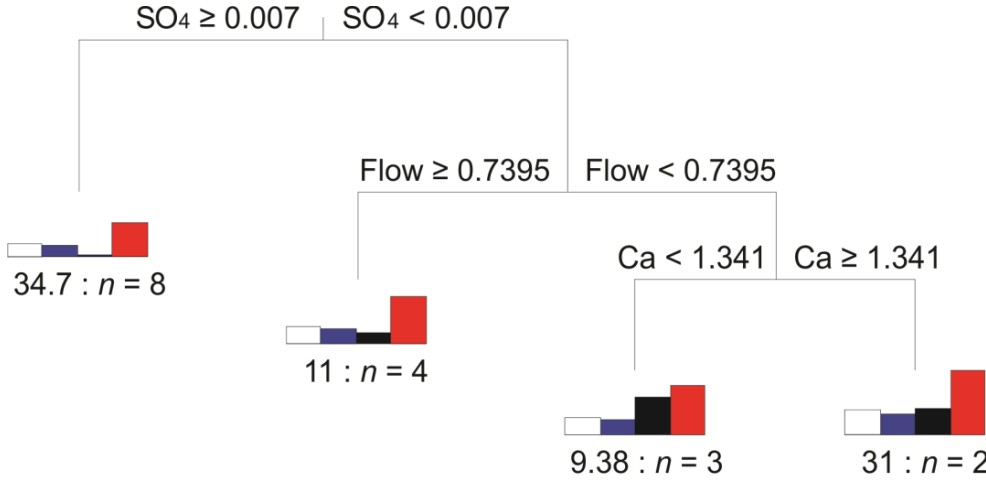

**Figure 4.** MRT for the biotic indices of Shannon (white), Simpson (blue), % EPT (black), and BMWP (red) using 48 different explanatory variables related to water quality, basin aspects, geomorphology, and habitat quality.

## 4. Discussion

The biotic integrity and water quality were defined according to the macroinvertebrate community structure and the river's environmental conditions at the in situ level and fluvial landscape. We analyzed physicochemical characteristics, geomorphological aspects, and habitat quality, as well as basin features to determine their relationship with the river ecosystem. We studied the main course of the La Pasión river and detected only a few sections with optimal water, habitat, and biotic conditions; most of the river presented different impacts related to economic activities and human settlements. Consequently, there is the discharge of domestic and industrial wastewater without treatment and the proximity of activities such as livestock and temporal and irrigation agriculture impose a diffuse impact on the river, an aspect better reflected by the water quality index (NFS-WQI). There are also modifications of the channel such as the canalization to take advantage of the

resource in irrigated agriculture and the rectification of the river in the urban area, aspects identified according to the suboptimal and marginal values of the habitat quality (S13 to S15). However, there are recreational areas with better water and biotic quality close to site S3, where spring water and two tributaries with clean waters enter the river (SR and S1).

The results obtained in the biological indices corroborate the first hypothesis proposed since there is a recovery of the quality of the river towards the outflow. This is related to the entry of clean water, where greater diversity is related to higher values of physicochemical parameters such as dissolved oxygen, water temperature, and transparency [59]. Different studies have described significant differences and important changes from the upper to the lower basin in the aquatic macroinvertebrate taxa, which could be related to changes in the physicochemical characteristics of the water by wastewater inputs, changes in habitats, and physical modifications to the channel, including nearby roads [60–62]. The values of the BMWP and the % EPT-B indices better reflected the general quality of the river, an aspect also described in other studies [45,63–65]. In terms of composition, families such as Polycentropodidae, Hydrophilidae, Philopotamidae, Helycopsychidae, Heptageniidae, Libellulidae, Sisyridae Ceratopogonidae, and Dugesiidae require high concentrations of dissolved oxygen and, therefore, were found in sites with good quality [5]. In contrast, some families of snails (Physidae and Planorbidae), Odonata (Coenagrionidae, Gomphidae, Hydroptilidae, Libellulidae, Calopterygidae, and Lestidae), the tolerant families of mayflies (Baetidae), and caddisflies (Limnephilidae, Hydropsychidae, and Polycentropodidae), as well as some beetles (Corixidae, Naucoridae, Noteridae, and Dytiscidae) are related to lentic habitats [66], with physical modifications in the riverbed and activities such as irrigated agriculture, as occurred in different sites located in the middle and lower basin [20,67].

Regarding landscape aspects, open areas related to agricultural land use impose ecological and physiological restrictions on the dispersal of species from families such as Diptera, Coleoptera, Ephemeroptera, and Trichoptera, whereas odonates are favored [20]. In addition, sites closely related to irrigated agriculture are characterized by high levels of electrical conductivity and nutrients (nitrogenous compounds, total phosphate, and sulphates), increased water temperature, increased chemical and biochemical oxygen demand, and a greater presence of suspended solids and organic matter, aspects that could induce eutrophication [68–70], thus affecting transparency, decreasing sunlight entry and primary production, as well as increasing turbidity. On the other hand, the heterogeneity and complexity of the habitats presents a greater richness [71,72]. The maximum biodiversity is maintained at intermediate disturbance and resource availability, levels typically found in conserved riverine areas or those with less anthropogenic influence and different substrate types [67,73]. The most frequent families with the largest number of individuals indicate the presence of organic pollution, mainly those related to the consumption of detritus such as Gammaridae, Baetidae, Chironomidae, Hydrachnidia, Asellidae, and Physidae, which were also abundant in altered ecosystems such as urban rivers [45,74]. In relation to habitat quality, the influence of this productive activity results in suboptimal to marginal results, which determine the presence of tolerant macroinvertebrate families [22].

A particular aspect is that while the results of the habitat characterization protocol provided results from suboptimal to marginal that were related to physical modifications (SCR), medium and good values can be found from the biotic quality due to the presence of aquatic plants, aspects related to morpho-group diversity and richness in rivers in Europe [75,76]. Aquatic plants, including introduced species such as the water hyacinth (*Eichhornia crassipes*), provide different habitats for macroinvertebrates; accordingly, sites that have poor water and habitat quality (e.g., S15) had high values in the BMWP index [46].

Regarding our second hypothesis, the analysis (MRT) only identified the main variables that describe the macroinvertebrates' structure and some parameters of the water and habitat quality. Although this is to be expected because they describe the environment where the organisms directly develop [46], they also indirectly reflect the impacts of the activities that occur at a larger scale around the river such as agricultural activities and (in

some other rivers) coal mining [77,78]. This has been described in other studies where the physicochemical variables in the water and the river characteristics are analyzed [5,79]

## 5. Conclusions

In this study we conclude with the results obtained in the water quality index, biological indices, and environmental and habitat characteristics that the spatial and temporal variation of aquatic macroinvertebrates is mainly affected by changes in the physicochemical characteristics in water due to the entry of domestic and industrial wastewater without prior treatment in the La Pasión river. A noticeable change is observed starting at site three (S3); however, at site 11 (S11) the most significant modification occurs in the river, an irrigation channel through which most of the water resource is captured for the irrigated agriculture area during the drier season. The relationship between environmental variables and macroinvertebrate diversity is a key point for understanding processes in aquatic ecosystems and adjacent activities and is the basis for evaluating possible methods for river restoration and pollution mitigation.

We show that different land use variables (lowland semi-deciduous forest, pasture, cultivated land, and human settlements), hydromorphological variables, environmental characteristics, and habitat quality have different impacts on different macroinvertebrate indicator groups at the family level. Specifically, we found that the distribution of macroinvertebrates such as EPT (a sensitive species) is clearly different from that of other macroinvertebrate families. However, the life cycle of aquatic organisms and their distribution in different habitats must be considered. Our finding is reinforced by the independent responses of species (individual families) to changes in environmental gradients.

**Author Contributions:** Investigation and writing—original draft, L.F.G.-S.; formal analysis, methodology and writing—review and editing, R.M.-E. and M.A.V.-M.; software and validation, G.C.-C.; resources and supervision, L.A.Á.-M. and J.L.P.-E. All authors have read and agreed to the published version of the manuscript.

**Funding:** This research was supported by the Instituto Politécnico Nacional (SIP 20211058).

**Data Availability Statement:** Not applicable.

**Acknowledgments:** Special thanks to CONACyT and the Instituto Politécnico Nacional for the financial support. We thank Estanislao Martínez-Bravo for his help with the hard field work.

**Conflicts of Interest:** The authors declare no conflict of interest.

## Abbreviations

| | |
|---|---|
| MLR | Machine learning regression algorithm |
| NDVI | Normalized difference vegetation index |
| NSF-WQI | National Sanitation Foundation index of water quality |
| EPT-B % | Percentage of Ephemeroptera, Plecoptera, and Trichoptera (minus Baetidae) |
| BMWP | Biological Monitoring Working Party index |
| NMDS | Non-metric multidimensional scaling |
| MRT | Multivariate regression tree analysis |
| HQI | Habitat quality index |
| B | Boron |
| BS | Bank Stability |
| CA | Channel alteration |
| CFS | Channel flow status |
| Emb | Embeddedness |
| FE | Frequency of riffles |
| IA | Irrigated agriculture |
| IP | Induced pastureland |
| LSF | Lowland semideciduous forest |
| RVZW | Riparian vegetative zone width |
| RA | Rainfed agriculture |

| | | |
|---|---|---|
| UZ | Urban zone | |
| VP | Vegetative protection | |
| V/DR | Velocity/depth regime | |

## Appendix A

**Table A1.** Physicochemical variables determined in the different sites of the La Pasión river.

| Sites | pH | EC (μS/cm) | Ca²⁺ (mg/L) | Mg²⁺ (mg/L) | Na⁺ (mg/L) | K⁺ (mg/L) | CO₃ (mg/L) | HCO₃ (mg/L) | Cl (mg/L) | SO₄²⁻ (mg/L) | B (mg/L) |
|---|---|---|---|---|---|---|---|---|---|---|---|
| S1 | 7.73 | 264.3 | 1.44 | 3.32 | 1.77 | 2.95 | 0.13 | 3.13 | 15.09 | 0.0002 | 0 |
| S2 | 7.98 | 423 | 1.49 | 2.28 | 2.02 | 3.33 | 0 | 4.20 | 26.41 | 0.011 | 0.249 |
| S3 | 8.24 | 502 | 1.64 | 2.25 | 2.12 | 4.48 | 0 | 4.85 | 33.95 | 0.008 | 0.21 |
| S4 | 6.87 | 1664 | 2.21 | 3.00 | 7.18 | 5.65 | 0 | 9.51 | 192.39 | 0.073 | 0.843 |
| S5 | 7.68 | 690 | 1.12 | 3.83 | 1.22 | 3.31 | 0 | 4.35 | 18.86 | 0.047 | 0 |
| S6 | 7.18 | 368 | 1.24 | 3.58 | 2.05 | 3.81 | 0 | 3.62 | 22.63 | 0.006 | 0 |
| S7 | 7.64 | 321 | 1.04 | 0.49 | 0.80 | 2.69 | 0 | 3.49 | 18.86 | 0.0003 | 0 |
| S8 | 7.65 | 340 | 1.69 | 0.74 | 1.05 | 3.82 | 0 | 3.54 | 18.86 | 0.002 | 0 |
| S9 | 7.65 | 347 | 1.32 | 1.36 | 2.47 | 3.27 | 0 | 3.56 | 18.86 | 0.0018 | 0 |
| S10 | 7.57 | 341 | 1.54 | 1.15 | 1.26 | 3.88 | 0 | 3.56 | 18.86 | 0.004 | 0 |
| S11 | 8.04 | 332 | 1.11 | 3.94 | 1.82 | 4.76 | 0.25 | 3.54 | 18.86 | 0.006 | 0 |
| S12 | 8.08 | 360.2 | 1.99 | 0.56 | 1.15 | 4.08 | 0 | 3.62 | 15.09 | 0.008 | 0 |
| S13 | 8.49 | 464 | 3.51 | 3.36 | 1.86 | 3.28 | 0.56 | 5.18 | 26.41 | 0.012 | 0.2235 |
| S14 | 8.29 | 546 | 3.93 | 3.88 | 2.89 | 3.83 | 0.33 | 5.59 | 30.18 | 0.015 | 0.275 |
| S15 | 7.71 | 1004 | 0.08 | 5.22 | 3.39 | 2.73 | 0 | 9.58 | 49.04 | 0.102 | 0.468 |
| SR | 7.73 | 264.3 | 1.44 | 3.32 | 1.77 | 2.95 | 0.13 | 3.13 | 15.09 | 0.0002 | 0 |
| SCR | 8.04 | 332 | 1.11 | 3.94 | 1.82 | 4.76 | 0.25 | 3.54 | 18.86 | 0.006 | 0 |

| Sites | NH₄ (mg/L) | PO₄³⁻TP (mg/L) | Hardness (mg/L CaCO₃) | NO₃ (mg/L) | COD (mg/L) | O and G (mg/L) | BOD₅ (mg/L) | TN (mg/L) | ET (°C) | WT (°C) | DOPS (%) |
|---|---|---|---|---|---|---|---|---|---|---|---|
| S1 | 4.71 | 0.12 | 100 | 0 | 12.33 | 0 | 2.12 | 4.71 | 17.7 | 25.8 | 95.9 |
| S2 | 1.17 | 1.23 | 115 | 0.04 | 15.67 | 0.25 | 2.7 | 1.21 | 18 | 25 | 61.7 |
| S3 | 1.42 | 0.55 | 160 | 0.32 | 149 | 0.04 | 27.21 | 1.74 | 17.3 | 17.9 | 47.3 |
| S4 | 2.38 | 25.87 | 340 | 2.16 | 1065.67 | 0.27 | 477.19 | 4.55 | 16.7 | 18.1 | 44.2 |
| S5 | 1.25 | 0.18 | 130 | 0 | 42.33 | 0 | 369.05 | 1.25 | 17.3 | 18.2 | 39.8 |
| S6 | 11.57 | 3.65 | 130 | 0.10 | 12.33 | 0 | 188.54 | 11.66 | 19.2 | 18.7 | 46.4 |
| S7 | 2.39 | 0.00 | 135 | 0.10 | 159 | 0 | 27.4 | 2.50 | 19.1 | 18.5 | 38.9 |
| S8 | 2.04 | 3.33 | 110 | 0 | 185.67 | 0.24 | 81.9 | 2.04 | 19.4 | 18.6 | 51.5 |
| S9 | 3.25 | 3.01 | 125 | 0.10 | 29 | 0.13 | 62.83 | 3.35 | 19 | 18.5 | 59.2 |
| S10 | 1.88 | 0.37 | 125 | 0 | 2.33 | 0.24 | 25.04 | 1.88 | 19.2 | 18.7 | 54.6 |
| S11 | 3.10 | 2.83 | 125 | 0.72 | 12.33 | 0.01 | 107.3 | 3.82 | 19.2 | 18.7 | 60.5 |
| S12 | 2.04 | 3.20 | 130 | 0.35 | 39 | 0 | 596.36 | 2.39 | 23.1 | 19.7 | 59.7 |
| S13 | 6.59 | 0.12 | 210 | 0.10 | 12.33 | 0 | 188.54 | 6.68 | 24.7 | 19.7 | 44.3 |
| S14 | 2.27 | 1.28 | 230 | 0.01 | 19 | 0.22 | 165.65 | 2.29 | 23.2 | 21 | 53.6 |
| S15 | 13.96 | 22.07 | 305 | 0.44 | 142.33 | 0.03 | 667.02 | 14.40 | 28.7 | 21.1 | 45.9 |
| SR | 4.71 | 0.12 | 100 | 0 | 12.33 | 0 | 2.12 | 4.71 | 17.7 | 25.8 | 95.9 |
| SCR | 3.10 | 2.83 | 125 | 0.72 | 12.33 | 0.01 | 107.3 | 3.82 | 19.2 | 18.7 | 60.5 |

| Sites | DO (mg/L) | Sal (‰) | Trans (cm) | Turb (NTU) | TDS (mg/L) | TotCol (MPN 100 mL) | FecCol (MPN 100 mL) | *E. coli* (MPN 100 mL) | Flow (m³/s) |
|---|---|---|---|---|---|---|---|---|---|
| S1 | 6.27 | 0.13 | 0.61 | 19 | 166.4 | 40 | 30 | 30 | 0.27 |
| S2 | 4.11 | 0.11 | 0.13 | 65 | 147.2 | 230 | 150 | 100 | 0.64 |
| S3 | 3.64 | 0.14 | 0.05 | 100 | 205.8 | 2100 | 430 | 110 | 0.69 |
| S4 | 3.39 | 0.14 | 0.25 | 50 | 205.8 | 930 | 280 | 40 | 0.70 |
| S5 | 3.04 | 0.13 | 0.11 | 50 | 198.45 | 1500 | 930 | 280 | 1.19 |
| S6 | 3.52 | 0.12 | 0.16 | 40 | 183.75 | 2400 | 1500 | 150 | 0.49 |
| S7 | 2.97 | 0.13 | 0.16 | 35 | 198.45 | 930 | 150 | 90 | 0.69 |
| S8 | 3.92 | 0.09 | 0.38 | 13 | 132.3 | 280 | 210 | 90 | 0.95 |
| S9 | 4.52 | 0.11 | 0.55 | 14 | 169.05 | 230 | 110 | 30 | 0.79 |
| S10 | 4.16 | 0.12 | 0.65 | 10 | 176.4 | 210 | 110 | 90 | 1.01 |
| S11 | 4.63 | 0.09 | 0.39 | 11 | 139.65 | 430 | 90 | 40 | 0.69 |
| S12 | 4.5 | 0.11 | 0.55 | 9 | 169.05 | 210 | 110 | 70 | 0.30 |
| S13 | 3.34 | 0.10 | 0.56 | 30 | 147 | 4600 | 1500 | 90 | 0.62 |
| S14 | 3.95 | 0.10 | 0.24 | 24 | 154.35 | 1500 | 230 | 110 | 0.51 |
| S15 | 3.38 | 0.11 | 0.13 | 24 | 169.05 | 2100 | 930 | 280 | 0.64 |
| SR | 6.27 | 0.13 | 0.93 | 19 | 166.4 | 40 | 30 | 30 | 0.27 |
| SCR | 4.63 | 0.09 | 0.54 | 11 | 139.65 | 430 | 90 | 40 | 6.97 |

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
