# Peer review of "Biotic Integrity, Water Quality, and Landscape Characteristics of a Subtropical River"

_water, doi:10.3390/w15091748_

Round 1

Reviewer 1 Report

1. The title is not good and should be revised.

2. No obvious results could be found in the abstract.

3. Introduction is no charming, I don't why this work was conducted, and no obvious hypothesis could be found.

4. Biotic integrity, water quality, and landscape characteristics should be combined together to analyze.

5. 

This manuscript could be accepted after major revision

Reviewer 2 Report

I think you have an interesting paper and I enjoyed reading it.  However, while indexes, WQI, HQI, EPT%, etc, contribute to distinguishing among sites.  In the file attached I've suggested that you include a table having a subset of the physiochemical parameters. A similar table addressing the macroinvertebrate data might also be helpful.

I have recommended that the editors undertake an English Language review for you. I did so because of many instances where I felt you had difficulty in describing your meaning or thoughts.  

Reviewer 3 Report

Review to Manuscript WATERS – 2346960

The article describes the environmental characteristics of a tropical watercourse by analysing benthic macroinvertebrate communities, the chemical characteristics of the water and the environmental structure of the surrounding areas. Its contents is not new however it provides useful information on the process of ecological transformation of a watercourse and may be an interesting point of comparison for the future.

Having said that, the paper presents several problems that must be addressed and solved before publication.

1) The article is too long and articulate for the information it provides: it must be be conveniently shortened by intervening on the methods part and combining results and discussion in a single paragraph

2) Summarising its content should also clear up some of the confusion that emerges from reading it and make the results more clear and sounding

3) The statistics applied seem disproportionate for the type of information they provide. The text could benefit from their simplification.

4) A review of English is strongly suggested.

More details in the following rows:

1 – Title: “Biotic integrity, water quality, and landscape characteristics of a subtropical river” its enough. How you do that can be read in the article.

2 – text from R69 to 76 flatty repeats flatly text from 64 to 68. Why?

3 – R81: insert Mexico

4- Figure 1. Insert Mexico in the figure caption. Try to separate Mexico general map from detailed map.

5-Paragraph 2.2: Describe how the sampling points were distributed in the study area (regularly, randomly or other)

6- R100: “The density values of the families were similar to other months within the season (December)”. What does it means? Are you talking of Population density of some particular species? Families do not show any “density”. Please clarify.

Moreover: at R99 you stated that river flow is at its low. Then, that organisms sampling is more effective (i.e. their abundance is higher). However, at R100, it seems you are saying that organism density is similar to those monitored in other months (when the river level is different, I suppose). Something is wrong. Clarify.

7- R133-138: unclear. In my opinion you should describe briefly the “criteria proposed by 59”, and “the habitat protocol proposed by 14”. In the Method paragraph the information from 59 and 14 are important to understand what you have done.

8- R147: the same as above. What are the infos by INEGI ?

9-R172. I suggest “Since data distribution significantly differed by normality (Shapiro test), non-parametric test as K-Wallis etc etc were adopted to test for differences between density in sampling sites”

10-R173: unclear. Must be rewritten

11-R197. I suggest to start from a Table or a Figure showing the habitat surface for any sample areas. “Habitat x y and z showed the higher surface, while w, h and j ………..

12-Figure 2. Unclear. First of all, numbers, words and any symbols along x and y axes must be easy to read (I suggest a 16 or 18 font size). Secondly, a legend explaining the meaning of acronyms below any histogram is strongly needed.

If S1, S2 etc are sampling point, is urgent to explain what the term “parameters” on y axes stay for. How many parameters? Which parameters? Why ther are more parameters and only a single bar in the histograms? If, instead, S2 S2 etc are parameters, numbers on y axes should be percentage or frequency.

13-R238: unclear. Are you saying that species richness and habitat diversity are significantly and positively related?

14- Figure 3 should be divided into two adjacent figures (Fig 3a and Fig 3b). In the first, you’ll show the habitat around the sampling areas. In the second, the spatial distribution of taxa, according to habitat. Only by doing this you can perceive a pattern in the distribution of organisms according to river parameters. The current Figure 3 as it stands is confused and unreadable.

15- In figure 4  why >= instead of ≥  ?

Need revisions

Reviewer 4 Report

The manuscript is devoted to the complex assessment of the state of the subtropical river La Pasion River with the use of Water Quality Index, Habitat Quality Index, Ephemeroptera, Plecoptera, Trichoptera Index minus Baetida and Biological Monitoring Working Party Index to describe the effect of human-induced activity on landscape and ecosystem characteristics. The authors demonstrated the noticeable influence of untreated domestic and industrial wastewaters discharges. The conclusions of authors are well documented, supported and illustrated by good graphic materials and tables. As an advice I’d recommend some more rooted in “fundamental ecology” variables, like ecological thermodynamics parameters: entropy, emergy, exergy etc., some of them where successfully applied to the assessment of rivers under wastewaters impact, as far as I can remember.

It is better to avoid the use of abbreviations, excluding widely known (like GIS, BOD, COD) in the text of abstract, in keywords, or figure captions and table headings, as they must be informative even taken apart from the main body of the manuscript. I’d propose to list all the abbreviations somewhere between keywords and introduction. There are some misprints and miswrites in the text, so it must be carefully checked. Nevertheless, the general impression from the manuscript – it is good job worse to be published.

Round 2

Reviewer 2 Report

Thank you for considering my suggestions for improving the earlier version of your manuscript. I believe your manuscript is improved and have no further suggestions.

The authors addressed all specific comments I made on English usage on an earlier version of their manuscript.

In a few instances I made comments on English usage that were not specific, consequently the authors were not able to respond. 

Author Response

We greatly appreciate the reviewer's comments and corrections that substantially enriched the manuscript. We additionally incorporated the observations from the other reviewers.

Reviewer 3 Report

Paper has been revised and is better than the first version. However it still presents some problems that need to be addressed and resolved before publication.

Basicly, some of my request has not been fulfilled and this prevent my “nulla-osta” to the publication.

1) The article is too long and articulate for the information it provides: it must be shortened by intervening on the methods part and combining results and discussion in a single paragraph

2) The statistics applied seem disproportionate for the type of information they provide. The text could benefit from their simplification.

The paper can be published after major revision.

DETAILS (see the first version)

5-Paragraph 2.2: Describe how the sampling points were distributed in the study area (regularly, randomly or other)

6- R100: “The density values of the families were similar to other months within the season (December)”. What does it means? Are you talking of Population density of some particular species? Families do not show any “density”. Please clarify.

Moreover: at R99 you stated that river flow is at its low. Then, that organisms sampling is more effective (i.e. their abundance is higher). However, at R100, it seems you are saying that organism density is similar to those monitored in other months (when the river level is different, I suppose). Something is wrong. Clarify.

7- R133-138: unclear. In my opinion you should describe briefly the “criteria proposed by 59”, and “the habitat protocol proposed by 14”. In the Method paragraph the information from 59 and 14 are important to understand what you have done.

8- R147: the same as above. What are the infos by INEGI ?

9-R172. I suggest “Since data distribution significantly differed by normality (Shapiro test), non-parametric test as K-Wallis etc etc were adopted to test for differences between density in sampling sites”

10-R173: unclear. Must be rewritten

11-R197. I suggest to start from a Table or a Figure showing the habitat surface for any sample areas. “Habitat x y and z showed the higher surface, while w, h and j ………..

12-Figure 2. Unclear. First of all, numbers, words and any symbols along x and y axes must be easy to read (I suggest a 16 or 18 font size). Secondly, a legend explaining the meaning of acronyms below any histogram is strongly needed.

If S1, S2 etc are sampling point, is urgent to explain what the term “parameters” on y axes stay for. How many parameters? Which parameters? Why ther are more parameters and only a single bar in the histograms? If, instead, S2 S2 etc are parameters, numbers on y axes should be percentage or frequency.

13-R238: unclear. Are you saying that species richness and habitat diversity are significantly and positively related?

14- Figure 3 should be divided into two adjacent figures (Fig 3a and Fig 3b). In the first, you’ll show the habitat around the sampling areas. In the second, the spatial distribution of taxa, according to habitat. Only by doing this you can perceive a pattern in the distribution of organisms according to river parameters. The current Figure 3 as it stands is confused and unreadable.

15- why "Turbidity >= 45" instead of "Turbidity  45" ?

Round 3

Reviewer 3 Report

The paper can be published.

My last suggestions, before publication:

R90-94: The study was carried out along the river and adjacent channels and at 18 sampling sites, regularly distributed throughout the study area, we measured several variables including natural features (e.g., tributaries inflow and river mouth), anthropogenic impacts (e.g., domestic, and industrial wastewater inputs). Areas with low or without impact were adopted as a control.

Figure 4 and Figure 4: please don't use >= or <=. Use instead  ≥ or 

.